# Temperature Changes during Er:YAG Laser Activation with the Side-Firing Spiral Endo Tip in Root Canal Treatment

**Sharonit Sahar-Helft** [1,2], **Nathanyel Sebbane** [1,3,4], **Adi Farber** [1,3], **Ronit Vogt Sionov** [3], **Roni Dakar** [2]
**and Doron Steinberg** [3,*]

1   Department of Endodontics, The Faculty of Dental Medicine, The Hebrew University-Hadassah, 9112102 Jerusalem, Israel; sharonit.sahar-helft@mail.huji.ac.il (S.S.-H.); nathanye.sebbane@mail.huji.ac.il (N.S.); adi.farber@mail.huji.ac.il (A.F.)
2   Endodontics Department, School of Graduate Dentistry, Rambam Health Care Campus, The Ruth and Bruce Rappaport Faculty of Medicine, Technion-Israel Institute of Technology, 3200003 Haifa, Israel; dr.ronidakar@gmail.com
3   Biofilm Research Laboratory, The Institute of Biomedical and Oral Research (IBOR), The Faculty of Dental Medicine, The Hebrew University of Jerusalem, 9112102 Jerusalem, Israel; ronit.sionov@mail.huji.ac.il
4   "Bina" Program, Faculty of Dental Medicine, The Hebrew University-Hadassah, 9112102 Jerusalem, Israel
*   Correspondence: dorons@ekmd.huji.ac.il; Tel.: +972-52-468-1999

**Abstract:** Erbium-doped yttrium aluminum garnet (Er:YAG) laser-activated irrigation (LAI) is used in endodontic treatment to remove the smear layer and kill bacteria in the root canal. However, this procedure can cause photo-thermal effects that harm the surrounding tissue. Therefore, it was important to study the temperature changes that occur at the outer tooth surface during activation of the Er:YAG laser using a side-firing spiral Endo tip in the root canal. Laser treatment was performed either in the absence of fluid in the root canal or in the presence of a 17% EDTA solution. Irrigation with 17% EDTA was either performed in a continuous mode for 60 s or in a segmented mode of 4 rinses with 17% EDTA for 15 s each. The temperature was measured every second during the treatment at three tooth surface sites: the cementoenamel junction, the middle region and the apical region. Our data show that the greatest temperature changes occurred when the laser was used alone without an irrigation solution, while minor temperature changes were observed with continuous irrigation. In conclusion, we would recommend applying the laser treatment with an irrigation solution to avoid excessive heating.

**Keywords:** Er:YAG laser; smear layer; bacteria; temperature; endodontic treatment; new side-firing spiral Endo tip

## 1. Introduction

The primary goal of root canal treatment (RCT) is to remove all microorganisms from the inner surface of the root canal system, prevent reinfection and establish healthy periapical tissue [1,2]. One approach is the biomechanical instrumentation of the root canal system, in which centrifugal forces are generated by the movement of the instrument and its proximity to the dentin wall, where a thicker layer called the smear layer is formed. The smear layer contains organic and inorganic substances, including remnants of odontoblastic processes, vital and necrotic pulp tissue, microorganisms and blood cells [3,4]. In 1975, McComb and Smith [5] were the first to describe the presence of the smear layer on the surface of an instrumented root. Mader et al. [6] discovered that the smear material is present in two layers: the outer superficial smear layer and the inner layer of materials densely packed in the dentinal tubules, reaching a depth of 40 μm [6]. When surface-active reagents are used, this depth can reach 110 μm [7]. Bacteria can survive [8] and proliferate in the dentinal tubules [9]. The smear layer clogs the dentinal tubules and prevents the penetration and activation of disinfectant irrigations [10]. Moreover, it can impair the sealing and adaptation of root canal obturation materials [11].

There are several methods of removing the smear layer. The most studied are chelating agents such as ethylenediaminetetraacetic acid (EDTA). Application of EDTA for 1–5 min is considered optimal. EDTA chelates calcium ions in dentin, resulting in soluble calcium chelates. Moodnik and Sulewski [12,13] showed that endodontic instruments' action and chemical irrigation cannot completely remove the smear layer from root canal walls. The conventional irrigation systems using NaOCl and EDTA solutions are unable to remove smear layers and organic debris in the apical third of the root canal [14]. Syringe irrigation is limited by the inability to extend the needle more than 1 mm from the tip [15]. The flushing is affected by the diameter of the canal, the cross-sectional shape of the canal, the depth of needle insertion and the diameter of the needle. The presence of air bubbles and vapor locks can further affect the efficiency of irrigation. These prevent the smooth flow of fluid into the narrow spaces of the fins, isthmuses and lateral canals. This is why neither conventional techniques such as hand or powered files nor irrigation routines can completely clean the root canal [16,17].

An innovative new modality of endodontic treatment combines the erbium–yttrium aluminum garnet (Er:YAG) laser with irrigants [18]. The physical effect of the laser on root canals depends on the absorption of its wavelengths in biological components and chromophores, such as water and apatite. Laser-activated irrigation (LAI) carried out by tools such as erbium lasers (Er:YAG/2980 nm and Er,Cr:YSGG/2780 nm) is becomes increasingly popular due to its effective removal of dentin smear layers and ability to disinfect root canals [19]. The use of laser irradiation in the root canal system has been shown to be effective in removing the smear layer and bacteria [20]. For this purpose, a side-firing spiral Endo tip was designed by Prof. Adam Shtabhols (The Hebrew University, Jerusalem, Israel) for specifically cleaning and disinfecting the root canal system during endodontic treatments and endodontic retreatments [20]. The tip is designed to fit the shape and volume of root canals prepared with the NiTi Protaper Gold (Dentspy Sirona, Charlotte, NC, USA) rotary instrument. The Endo tip is designed with a flexible, hollow, conical and round cross-section with circumferential spiral slits along its entire length (Figure 1). The end of the Endo tip is sealed, which prevents radiation transmission through the apical foramen. This unique design allows for efficient cleaning and disinfection. The tip is 25 mm long and has three zones: 1. the cylindrical sleeve zone (4 mm), which is inserted into the handpiece; 2. the flexibility zone (3 mm), which allows additional flexibility when the tip is in action in the root canal and 3. the functional area (18 mm) with six spiral slits. The width of each slit and the distance between them varies so that the width of the furthest apical side slit is narrower than the width of the slit located on the coronal (handpiece) side.

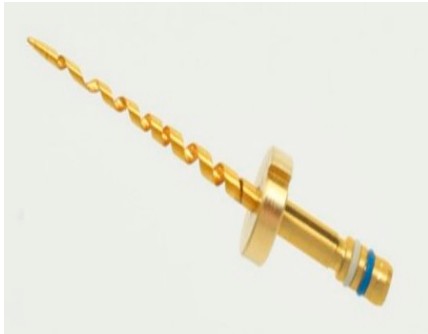
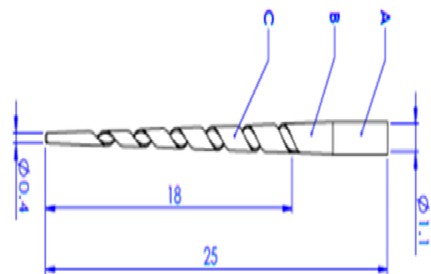

**Figure 1.** An image of the side-firing spiral Endo tip. The tip is hollow, conical and has a round cross-section with circumferential spiral slits along its entire length. It has three zones: A. the cylindrical sleeve zone (4 mm), which is inserted into the handpiece; B. the flexibility zone (3 mm), which allows additional flexibility when the tip is in action in the root canal; and C. the functional area (18 mm) with six spiral slits. The width of each slit and the distance between them varies so that the width of the furthest apical side slit is narrower (0.4 mm) than the width of the slit located on the coronal side (1.1 mm). The end of the Endo tip is sealed.

LAI primarily operates via cavitation shock, which causes the formation of vapor bubbles due to the absorption of laser energy by the liquid irrigation solution. These robust cavitation bubbles in the liquid enlarge during the pulse and then collapse inward, producing shock waves that lead to cleaning and disinfection of the root canal [21].

During root canal treatment, using an Er:YAG laser beam to activate the irrigation solution results in photo-thermal, photo-chemical and photo-ablative effects. This process can lead to a rise in the temperature and potentially damage the surrounding tissue, including the periodontal and bone areas. To avoid thermal damage to surrounding tissues, it is crucial to establish a safety limit for laser activation [22].

In 1983, Eriksson and Albrektsson determined a temperature of 47 °C as the critical limit for bone survival in rabbits [22]. In later studies, a temperature rise of 10 °C above body temperature was found to be the critical limit [23]. According to a 2001 dissertation by Mazaheri in BioMed Research International 7 at RWTH Aachen College, the maximum average temperature occurring during irradiation of root canals with a 3 W diode laser would fall below the critical limit if the optical fiber was continuously moved in a circular motion both coronally and apically [24]. Gutknecht et al. found that at a 3 W CW setting, bacterial reduction was observed in bovine teeth at a depth of 500 microns [25]. However, the temperature limit was exceeded at 4 W with an irradiation time of 15 s, resulting in thermal damage [25].

Gutknecht observed that using lasers below 1 W is irrelevant in endodontics because they do not completely remove the smear layer or seal the dentinal tubules [26]. However, at settings of 1.25 W to 1.5 W, the organic material is entirely removed, and the surface of the inorganic substance is fused [26]. These observations are important as they help define the conditions required for preventing reinfection in the root canal.

The present study aimed to examine the temperature changes occurring at the outer tooth surface during activation of the Er:YAG laser using the new side-firing spiral Endo tip in the root canal under the following treatment conditions: laser activation without fluid in the root canal and laser activation in the presence of a 17% EDTA solution in a continuous or segmented mode.

## 2. Materials and Methods

### 2.1. Tooth Preparation

Thirty single-rooted human teeth (central incisors) extracted for periodontal reasons were used in this study. The study was conducted with the approval of the Hadassah Hospital Ethics Committee, No. 0118-14-HMO.

The average working length of the root canal was 16.69 mm. After enlarging the canal openings with Gates-Glidden, the root canals were mechanically prepared with ProTaper files, similar to the routine clinical procedure. The root canals were enlarged to the apex with five different files for 10 cycles. The endodontic procedure was completed with file No. 40. The canal was washed with 1 mL of 2.5% sodium hypochlorite between the steps.

### 2.2. Laser Specification with the Side-Firing Spiral Tip for Endodontics

An Er:YAG laser that delivers a powerful 2940 nm wavelength (Light Instruments, Yokne'am, Israel) was used in this study. The energy used was 150 mJ, 10 Hz, for 60 s. According to the data obtained, a side-firing spiral Endo tip was used [27]. The Endo tip was inserted into the root canal to cover the entire area. The tip was moved up and down in the coronal–apical direction at 1–2 mm intervals.

### 2.3. Treatment Groups

Thirty single-rooted teeth were divided into three categories of ten teeth each:

Group A: laser irradiation for 60 s without fluid in the root canal.

Group B: laser irradiation for 60 s combined with continuous rinsing of the root canal with 10 mL of a 17% EDTA irrigation solution (DSI—Dental Solutions Israel Ltd., Ashdod, Israel).

Group C: four segments of laser irradiation for 15 s (total of 60 s), each cycle with 1 mL of a 17% EDTA irrigation solution in the root canal (total of 4 mL).

### 2.4. Temperature Measurements

The temperature rise at the outer tooth surface was recorded during laser activation when the Endo tip was operated inside the root canal. The temperature rise was measured using a digital readout thermometer coupled to a surface measurement probe (DAQPRO, Fourtec, Fourier Technology, Chatswood, NSW, Australia) with a sensitivity of 0.10 °C. The thermocouple was placed directly on the tooth surface at the cementoenamel junction (coronal area), in the middle of the root canal and at the apical area, and the temperature was recorded at 1 s intervals during the laser activity in the root canal (Figure 2). Each measurement was performed by the same operator for each source evaluation. The data collected through the computer and Microsoft Excel software were used to calculate the mean temperature rise, the range and the standard deviation for each tooth group. A statistical analysis of variance was used to determine the significance between the three treatment groups. The temperature was measured for 60 s each time; a total of 5400 s of temperature data was collected.

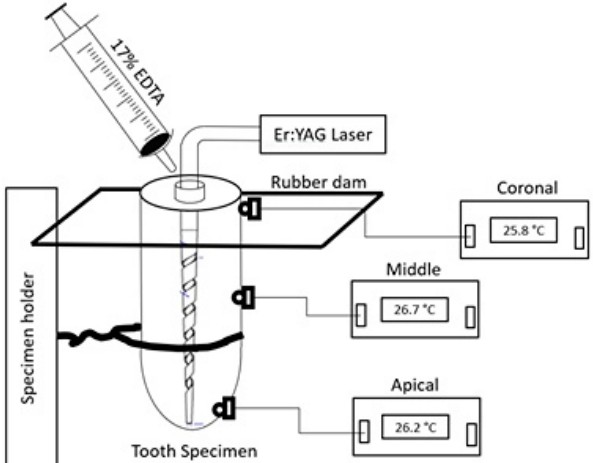

**Figure 2.** Illustration of temperature measurements. The image shows the Endo tip inserted into the root canal of a tooth specimen which is fixed in a specimen holder. The blue lines present laser firing. A 17% EDTA solution was injected to the canal simultaneously with the activation of the Er:YAG laser. The locations of the temperature measurements (coronal, middle and apical) are shown.

### 2.5. Statistical Analysis

ANOVA and post hoc Tukey's HSD test were used to determine the statistical difference in the temperature changes at the outer surface of the root canal between the three treatment groups as well as the temperature changes between the three areas (cementoenamel junction, middle area and apical area) within the same group. The statistical analysis was conducted using JUSP computer software (JASP Team (2022), Version 0.16.2). A *p*-value below 0.05 was considered statistically significant.

## 3. Results

### 3.1. Comparison of the Temperature Changes during Each Treatment Procedure

We compared the temperature changes ($\Delta T$) at the cementoenamel junction (coronal region), in the middle and apical regions of the tooth surface during laser treatment in the root canal under the following conditions: laser alone without irrigation solution (Control–Group A); laser in combination with continuous 17% EDTA irrigation (Group B); and laser in combination with four discontinued (segmented) 17% EDTA irrigations for 15 s each (Group C), resulting in a total treatment time of 60 s (Figure 3).

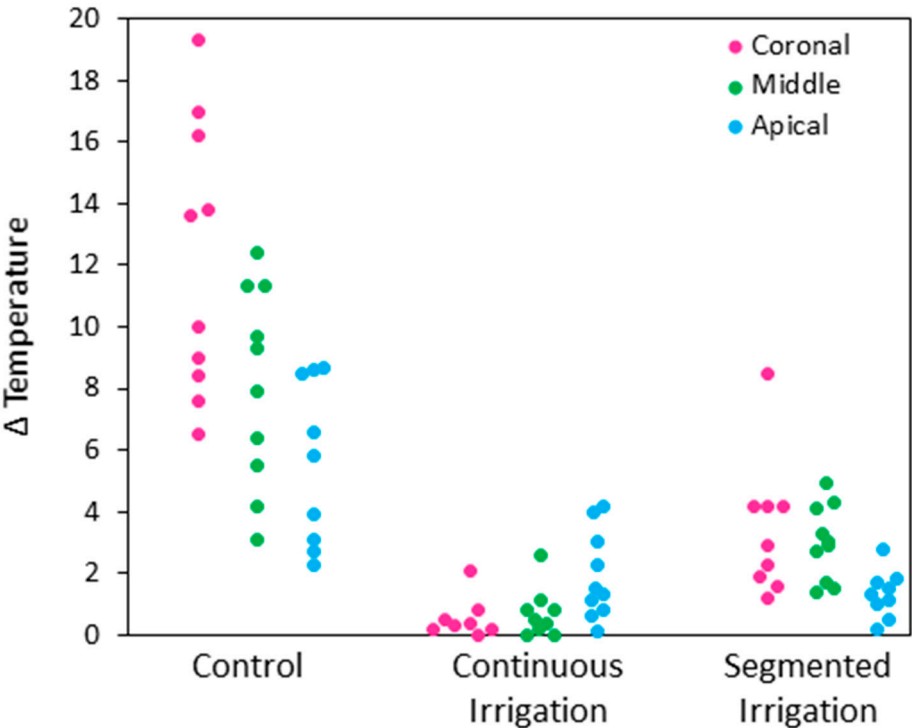

**Figure 3.** Temperature measurements of the outer tooth surface at the indicated locations (coronal, middle, apical) during the 60 s activation of the Er:YAG laser with the side-firing spiral Endo tip in the root canal for the three treatment conditions: Control was laser treatment without fluid in the root canal. Continuous irradiation was laser with 17% EDTA irrigation solution for 60 s. Segmented irritation was performed 4 times applying the laser with 17% EDTA irrigation solution for 15 s per cycle. Each circle represents the ΔT (Δ Temperature) of one sample. The pink circles represent the ΔT measurements at the coronal (cementoenamel junction) region. The green circles represent the ΔT measurements in the middle part, while the blue circles represent the ΔT measurements in the apical region.

The control group, which received the laser alone without irrigation solution (Group A), showed the greatest temperature changes, which were significantly higher ($p < 0.01$) than in the continuous (Group B) and segmented irrigation mode (Group C) (Figure 3 and Table 1). Minor temperature changes were observed in the continuous irrigation method (Group B), although there was no significant difference with the segmented method (Group C) (Figures 3–6 and Table 1).

**Table 1.** Post hoc Tukey's comparisons show statistically significant differences between the control Group A and the irrigation Groups B and C.

| Group | Compared to | Mean Difference | SE | *t* | $P_{\text{Tukey}}$ |
|---|---|---|---|---|---|
| Continued Irrigation | Control | 7.794 | 0.652 | −11.948 | <0.001 |
| Continued Irrigation | Segmented Irrigation | 1.526 | 0.664 | −2.300 | 0.062 |
| Control | Segmented Irrigation | 6.268 | 0.645 | 9.715 | <0.001 |

Notes: *p*-value is adjusted for comparing a family of 3. Results are averaged over the levels of Measrement_Point. Mean difference: average ΔT of the group of interest—average ΔT of the compared group. Standard error (SE) = the common error factor for all three groups. *t* = the sample statistical value under the T distribution. The $P_{\text{Tukey}}$ value stands for probability and measures the likelihood that any observed difference between groups is due to chance.

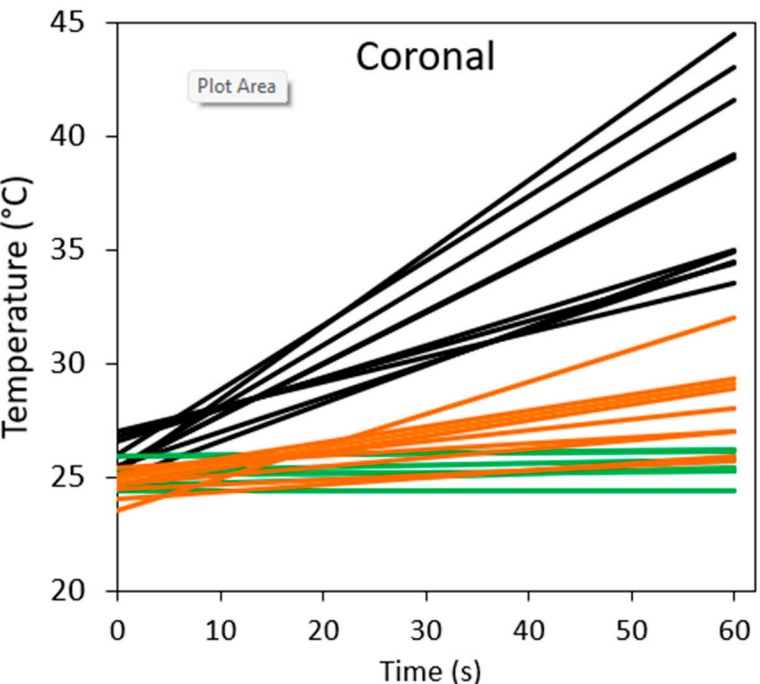

**Figure 4.** Temperature measurements of the outer tooth surface at the coronal site during the 60 s Er:YAG laser activation with the side-firing spiral Endo tip in the root canal for the three treatment conditions: Control is laser treatment without fluid in the root canal (black lines). Continuous irradiation was laser with 17% EDTA irrigation solution for 60 s (green lines). Segmented irritation was performed 4 times applying the laser with 17% EDTA irrigation solution for 15 s per cycle (orange lines). Each line is a separate sample.

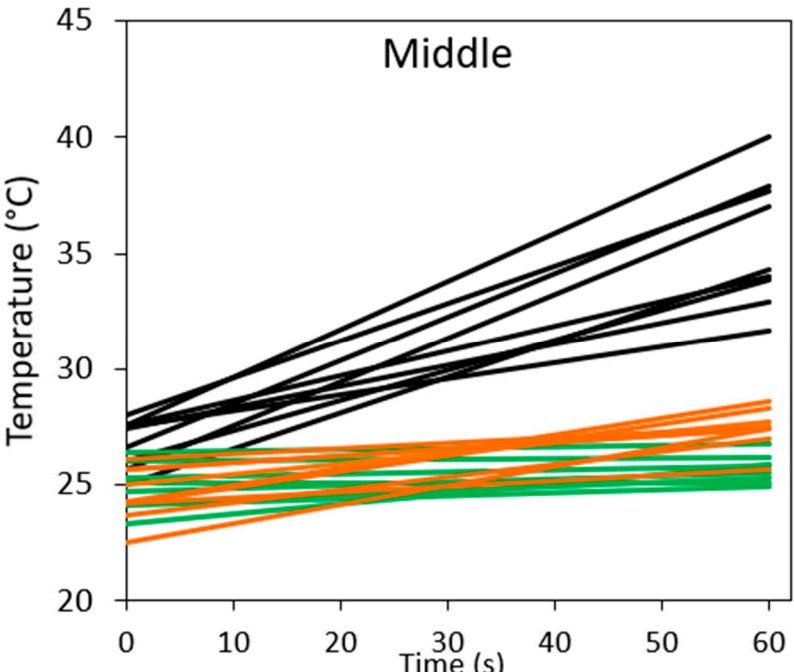

**Figure 5.** Temperature measurements of the outer tooth surface at the middle region during the 60 s Er:YAG laser activation with the side-firing spiral Endo tip in the root canal for the three treatment conditions: Control is laser treatment without fluid in the root canal (black lines). The continuous irradiation is laser with 17% EDTA irrigation solution for 60 s (green lines). The segmented irritation was performed 4 times applying the laser with 17% EDTA irrigation solution for 15 s per cycle (orange lines). Each line is a separate sample.

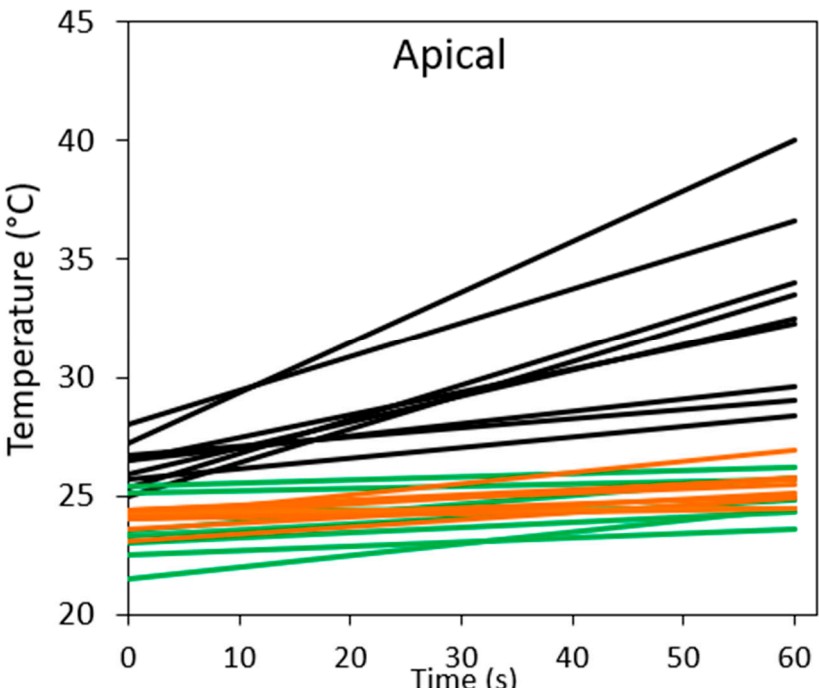

**Figure 6.** Temperature measurements of the outer tooth surface at the apical region during the 60 s Er:YAG laser activation with the side-firing spiral Endo tip in the root canal for the three treatment conditions: Control is laser treatment without fluid in the root canal (black lines). Continuous irradiation is laser with 17% EDTA irrigation solution for 60 s (green lines). Segmented irritation was performed 4 times applying the laser with 17% EDTA irrigation solution for 15 s per cycle (orange lines). Each line is a separate sample.

Results of the analyses of the statistical significance of the different temperature changes (ΔT) observed in the three treatment groups and at the three surface sites (cementoenamel junction, middle and apical area) are presented in Tables 1–3.

**Table 2.** Post hoc Tukey's comparisons showing statistically significant differences between the apical and coronal regions.

| Group | Compared to | Mean Difference | SE | $t$ | $P_{\text{Tukey}}$ |
|---|---|---|---|---|---|
| Apical | Coronal | −2.210 | 0.658 | −3.359 | <0.003 |
| Apical | Middle | −0.762 | 0.645 | −1.181 | 0.468 |
| Coronal | Middle | 1.449 | 0.658 | 2.202 | 0.077 |

Notes: $p$-value is adjusted for comparing a family of 3. Results are averaged over the levels of Measrement_Point. Mean difference: average ΔT of the group of interest—average ΔT of the compared group. Standard error (SE) = the common error factor for all three groups. $t$ = the sample statistical value under the T distribution. The $P_{\text{Tukey}}$ value stands for probability and measures the likelihood that any observed difference between groups is due to chance.

Post hoc Tukey's HSD test for multiple comparisons showed that the mean ΔT values significantly differed between the three groups. Table 1 presents the ΔT and the statistical significance between the different treatment methods: the control group of laser treatment without fluid, the continuous 17% EDTA irrigation method with the Er:YAG laser and the segmented 17% EDTA irrigation method with the Er:YAG laser. Table 2 presents the mean values of ΔT in different measurement areas (cementoenamel junction, middle and apical) and the statistical significance. Table 3 shows the treatment method, measurement points and statistical significance ($p < 0.001$ for irrigation groups versus control; Table 3). Moreover, the mean values of ΔT between the control and other groups were significantly different ($p < 0.001$ for irrigation groups versus control; Table 3).

**Table 3.** Post hoc Tukey's comparisons between the different treatment methods and measurement locations.

| Method | Surface Site | Compared to Method | Surface Site | Mean Difference | SE | t | $P_{\text{Tukey}}$ |
|---|---|---|---|---|---|---|---|
| Continued | Apical | Control | Apical | −4.406 | 1.097 | −4.015 | 0.004 |
| | | Segmented | Apical | 0.572 | 1.127 | 0.507 | 1.000 |
| | | Continued | Coronal | 1.331 | 1.164 | 1.144 | 0.965 |
| | | Control | Coronal | −10.246 | 1.097 | −9.337 | <0.001 |
| | | Segmented | Coronal | −1.550 | 1.127 | −1.375 | 0.904 |
| | | Continued | Middle | 1.183 | 1.127 | 1.049 | 0.979 |
| | | Control | Middle | −6.216 | 1.097 | −5.665 | <0.001 |
| | | Segmented | Middle | −1.086 | 1.097 | −0.990 | 0.985 |
| Control | Apical | Segmented | Apical | 4.978 | 1.127 | 4.415 | 0.001 |
| | | Continued | Coronal | 5.737 | 1.164 | 4.930 | <0.001 |
| | | Control | Coronal | −5.840 | 1.097 | −5.322 | <0.001 |
| | | Segmented | Coronal | 2.856 | 1.127 | 2.533 | 0.233 |
| | | Continued | Middle | 5.589 | 1.127 | 4.957 | <0.00 |
| | | Control | Middle | −1.810 | 1.097 | −1.649 | 0.774 |
| | | Segmented | Middle | 3.320 | 1.097 | 3.026 | 0.077 |
| Segmented | Apical | Continued | Coronal | 0.760 | 1.192 | 0.637 | 0.999 |
| | | Control | Coronal | −10.818 | 1.127 | −9.595 | <0.001 |
| | | Segmented | Coronal | −2.122 | 1.157 | −1.835 | 0.659 |
| | | Continued | Middle | 0.611 | 1.157 | 0.528 | 1.000 |
| | | Control | Middle | −6.788 | 1.127 | −6.021 | <0.001 |
| | | Segmented | Middle | −1.658 | 1.127 | −1.470 | 0.865 |
| Continued | Coronal | Control | Coronal | −11.577 | 1.164 | −9.947 | <0.001 |
| | | Segmented | Coronal | −2.882 | 1.192 | −2.417 | 0.290 |
| | | Continued | Middle | −0.149 | 1.192 | −0.125 | 1.000 |
| | | Control | Middle | −7.547 | 1.164 | −6.485 | <0.001 |
| | | Segmented | Middle | −2.417 | 1.164 | −2.077 | 0.496 |
| Control | Coronal | Segmented | Coronal | 8.696 | 1.127 | 7.713 | <0.001 |
| | | Continued | Middle | 11.429 | 1.127 | 10.138 | <0.001 |
| | | Control | Middle | 4.030 | 1.097 | 3.673 | 0.013 |
| | | Segmented | Middle | 9.160 | 1.097 | 8.348 | <0.001 |
| Segmented | Coronal | Continued | Middle | 2.733 | 1.157 | 2.363 | 0.319 |
| | | Control | Middle | −4.666 | 1.127 | −4.138 | 0.003 |
| | | Segmented | Middle | 0.464 | 1.127 | 0.412 | 1.000 |
| Continued | Middle | Control | Middle | −7.399 | 1.127 | −6.563 | <0.001 |
| | | Segmented | Middle | −2.269 | 1.127 | −2.013 | 0.540 |
| Control | Middle | Segmented | Middle | 5.130 | 1.097 | 4.675 | <0.001 |

Notes: *p*-value is adjusted for comparing a family of 3. Results are averaged over the levels of Measrement_Point. Mean difference: average ΔT of the group of interest—average ΔT of the compared group. Standard error (SE) = The common error factor for all three groups. *t* = the sample statistical value under the T distribution. The $P_{\text{Tukey}}$ value stands for probability and measures the likelihood that any observed difference between groups is due to chance.

### 3.2. Temperature Changes in the Coronal, Middle and Apical Tooth Sites

In the coronal region, the lowest temperature changes (ΔT) were measured during the continuous mode, with a median temperature increase of $0.56 \pm 0.67$ °C compared to $3.44 \pm 2.22$ °C using the segmented procedure (Figures 3 and 4). The highest temperature changes (ΔT) ($12.14 \pm 4.43$ °C) were measured in the laser method without irrigation solution in the root canal (Figures 3 and 4 and Tables 2 and 3).

In the middle region of the root canal, the following temperature changes (ΔT) were measured: $0.71 \pm 0.80$ °C for the continuous irrigation method; $2.98 \pm 1.21$°C for the segmented method; and $8.11 \pm 3.12$ °C for the control method without irrigation solution (Figures 3 and 5 and Tables 2 and 3).

At the apical area of the root canal, the following temperature changes (ΔT) were measured: $1.89 \pm 1.42$ °C for the continuous irrigation method; $1.32 \pm 0.76$ °C for the

segmented method; and 6.30 ± 3.39 °C for the control method without irrigation solution in the root canal (Figures 3 and 6 and Tables 2 and 3).

## 4. Discussion

The most important endodontic treatment is the removal of microorganisms and debris from the root canal system. Er:YAG LAI can achieve this effect through the potential energy of the light beam, which is strongly absorbed by the liquid. The cleaning mechanism involves cavitation bubbles that generate shock waves [28,29]. Due to the closed space of the root canals, there is a reduced bubble effect which results in a diminished shock waves effect [28]. To overcome the obstacle, Gregorcic et al. [29] investigated the possibility of increasing the laser effect using a liquid reservoir. The results show an acoustic effect that could be achieved by applying another laser pulse shortly after the first beam in the liquid reservoir. The collapse of the cavitation bubble increases the mechanical energy of the second oscillation [30]. However, in the limited root canal space, the laser light can only generate cavitation bubbles 2–3 mm near the fiber tip, so the liquid reservoir effect in the root canal is attenuated. Our study used the side-firing spiral Endo tip in root canal treatment. This unique design of the Endo tip allows the proximity of the laser beam to all parts of the canal walls. The laser generates efficient shock waves and oscillating fluid movement directly to the canal walls. With the Endo tip, we could efficiently remove the smear layer and bacterial biofilm and improve the cleaning mechanism inside the root canal [27].

There are two major challenges in root canal treatment. The first is the limited ability to clean the root canal using conventional methods and technologies. The second is the need to overcome obstacles such as anatomical complexities: dentinal tubules, lateral canals, isthmus and ramifications. Chemo-mechanical procedures include rotary and hand files with an irrigation solution. The traditional way of delivering the irrigating solution into the root canal is with a syringe and a 27-gauge or 30-gauge needle [31]. However, Fraser [32] found that the chelating effect was almost negligible in the apical third of the root canals. The irrigation solution can only reach 1–3 mm beyond the needle tip, depending on the needle type and irrigation flow. Several methods have been introduced in recent years to irrigate root canals better. We investigated the efficiency of different methods in our endodontic treatment [20]. Those methods included irrigation with 17% EDTA with positive pressure, passive ultrasound and laser-activated irrigations. The results show that using Er:YAG LAI laser with a 17% EDTA solution provided the best performance [20]. According to our studies, the proximity of the laser beam to the root canal wall is crucial. Sebbane et al. [27] showed that using an Er:YAG Endo tip with 17% EDTA effectively removed the biofilm from the entire root canal.

The present research focused on how the laser Endo tip affects the temperature in its vicinity. When the liquid absorbs the laser beam, it is transformed into heat, which generates internal pressure and vaporization, ultimately removing unwanted material from the root canal. Using the Er:YAG laser can cause thermal changes in the outer surface of the root canal, which may cause harm to the periodontal ligament and bone. Bahcall et al. [33] found that the use of a neodymium-doped yttrium aluminum garnet (Nd: YAG) laser in canine root canal treatment resulted in ankylosis, cementum lysis and significant bone remodeling 30 days after treatment. The potential temperature effects on surrounding structures are the primary concern when using lasers for root canal treatment. It is generally accepted that a temperature rise of about 10 °C above the average body temperature is considered critical [22]. Therefore, it is essential to analyze the temperature changes on the outer tooth surface following LAI, to ensure that the surrounding tissues are not damaged.

The temperature may vary between different segments of the tooth due to differences such as conductivity, depth and density. Therefore, we measured the temperature at various locations. When irrigation was applied with the laser, the temperature changes between the coronal, middle and apical zones were similar. The conductivity, material properties and irrigation minimized the temperature changes at the different locations of the tooth surface.

We investigated the thermal effect of the Er:YAG laser with the Endo tip on the outer surface of the root canal at three sites using three methods. The first method was a laser treatment without irrigation solution inside the root canal. The second approach was a continuous injection of a 17% EDTA irrigation solution into the root channel during laser activation for 60 s. In the third method, the 17% EDTA irrigation solution was applied four times as a reservoir into the pulp chamber for 15 s each time, for a total of 60 s of treatment of the root canal.

The results of our study show a significantly higher temperature increase ($12.14 \pm 4.43$ °C) can be achieved by activating the Er:YAG laser without liquid inside the root canal in the coronal part (Group A; $p < 0.01$) compared to the continuous (Group B) and segmented irrigation methods (Group C) (Figure 3), in which the temperature increased by only 0.5–2 °C on average. Minor temperature changes were observed in the continuous irrigation method (Group B). The $\Delta T$ in the continuous mode and the segmented method was lower than the safety limit of 10 °C. Therefore, using the Er:YAG laser with the Endo tip in root canal treatment is probably safe.

Using the continuous irrigation method (Group B), we observed that the temperature increased more in the apical region ($\Delta T$ of $1.89 \pm 1.42$ °C) than in the coronal ($\Delta T$ of $0.56 \pm 0.66$ °C) and middle regions ($\Delta T$ of $0.71 \pm 0.80$ °C) ($p = 0.02$). The explanation for this result could lie in the morphology of the root canal. The dentin structure is thinner in the apical part than in the coronal region. The results are similar to those observed by Kimura et al. [34]. Altogether, our findings demonstrate only a minimal thermal effect on the outer surface of the root canal during activation of the Er:YAG laser with the Endo tip when applied at 150 mJ, 10 Hz, for 60 s, along with 17% EDTA irrigation during root canal treatment. These parameters were chosen as they have been shown to provide good antibacterial and antibiofilm effects [27]. Other settings might result in a higher temperature of dentin tissue [35].

The purpose of this ex vivo study was to simulate the oral cavity of a patient. Although it is an ex vivo model, it is expected that the results can be translated into clinical trials and that the in vivo results in patients would be similar to our findings. The same laser properties, parameters and irrigation solutions used in the dental clinic were applied on the extracted teeth to mimic actual clinical conditions as much as possible. However, it is important to note that the clinical results may differ from those achieved with the ex vivo model due to variable environmental factors. For example, the teeth we used were extracted, not fresh live teeth, and the clinic's body temperature and room temperature may influence the in vivo results. Patients of different ages, oral hygiene and cultural backgrounds are treated in clinical practice. These factors may influence the size of the root canal, dentin tubules and the dentin and enamel structures. Using randomly extracted teeth from different patients, we found that the results were unaffected by the tooth diversity. Other variables that can differ from clinic to clinic are the temperature of the irrigation solution and the suction conditions, both of which can impact the $\Delta T$. Nevertheless, when used correctly, lasers are considered safe for ablating dental hard tissue [36].

Many different laser machines and Endo tips are available on the dental market. It is important to mention that an endodontics specialist in laser treatment conducted this study. A special training session on the use of lasers in the dental field is required to perform this kind of endodontic treatment. Given these efforts, it is likely that the results obtained in our study would be representative of what would also be observed in real clinical practice. Therefore, this study is a first step toward animal and subsequently human studies.

## 5. Conclusions

When the Er:YAG laser Endo tip was applied in the root canal without any fluid, there was a significant increase in the temperature (6–12 °C), which is intolerable from a clinical point of view. However, this temperature increase was prevented when conducting simultaneous irrigation in the root canal, with a minimal temperature change of 0.5–2 °C. Thus,

this study recommends that the laser treatment be performed with an irrigation solution to optimally clean the root canal and avoid thermal damage to the surrounding tissue.

**Author Contributions:** Data curation: S.S.-H., N.S. and A.F.; formal analysis: S.S.-H.; investigation: N.S. and A.F.; resources: S.S.-H. and D.S.; supervision: D.S., S.S.-H. and R.V.S.; writing—original draft: N.S., A.F. and S.S.-H.; writing—review and editing: R.D., D.S., S.S.-H., R.V.S. and N.S. All authors have read and agreed to the published version of the manuscript.

**Funding:** This study was partially funded by the Cabakoff Foundation, grant number 3175006340.

**Institutional Review Board Statement:** Not applicable.

**Informed Consent Statement:** Not applicable.

**Data Availability Statement:** Not applicable.

**Acknowledgments:** We thank Light Instruments, Yokne'am, Israel, for supplying the laser machine and the spiral Endo tip.

**Conflicts of Interest:** The authors declare no conflict of interest.

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
