# Peer review of "Temperature Changes during Er:YAG Laser Activation with the Side-Firing Spiral Endo Tip in Root Canal Treatment"

_photonics, doi:10.3390/photonics10050488_

Round 1

Reviewer 1 Report

This study aims to examine the temperature changes occurring at the outer canal surface during the activation of Er:YAG laser with the Endo tip side-firing spiral tip in the root canal. The research is interesting, However, There are some errors in the manuscript. For example, the background should not be included in the abstract part; the ordinate" Δ Temperature " should be standardized.

Author Response

We thank the reviewer for critically reading and commenting on our manuscript.

This study aims to examine the temperature changes occurring at the outer canal surface during the activation of Er:YAG laser with the Endo tip side-firing spiral tip in the root canal. The research is interesting, However, There are some errors in the manuscript.

  1. For example, the background should not be included in the abstract part.

We have accordingly rewritten the abstract.

  1. The ordinate" Δ Temperature " should be standardized.

We have accordingly corrected to ΔT throughout the text.

Reviewer 2 Report

The draft of the article titled "Temperature changes during Er: YAG laser activation with side-firing spiral Endo Tip in root canal treatment", shows the cleaning procedure to remove micro-organisms, as well as the use of laser for endodontic treatment, using a Er:YAG laser. The results shown can be interesting, however, there is a lack of explanation of what is happening with the determined temperature and its relationship with the application time and the response of the tissue. In this regard, I have the following comments:

1.     The document requires a complete revision of the writing, as well as the use of punctuation, for example, a punctuation symbol is missing on line 45, and line 75.

2.     What software, and data acquisition hardware were used by the authors to capture and record the different temperatures at a precise time?

3.     What is the meaning of the horizontal separation between the dots of the same treatment condition, i.e., for continuous irrigation, there are pink dots sparced in the axis, whit almost no temperature change.

4.     Figures 3, 4, and 5, show the same information as figure 2, I suggest eliminating those figures, or they could remain if there is a real reason to be there.

5. Tables 1, 2, and 3, are pictures with very bad quality, and their font size is different, table 3 is not easy to read. Authors must construct the tables from the latex editor, or the word editor, using a font with legible size.

6.     Results shown in table 3 are not explained in the document, a very good way to explain it is by using figures that can be described in the text. Also, table 3 doesn’t show the results of the segment.

7.     A figure, or figures, showing the temperature change versus the irrigation time for the different experiments is mandatory for a better explanation of the obtained results. 

I think that the analysis carried out, as well as the results obtained and the explanation of these, are very poor, and the document requires a lot of work, to be considered acceptable for publication. In this case, I will allow the work to be improved and suggest major comments, hoping that the work will be substantially improved in the next version.

Author Response

We thank the reviewer for critically reading and commenting on our manuscript.

The draft of the article titled "Temperature changes during Er: YAG laser activation with side-firing spiral Endo Tip in root canal treatment", shows the cleaning procedure to remove micro-organisms, as well as the use of laser for endodontic treatment, using a Er:YAG laser. The results shown can be interesting, however, there is a lack of explanation of what is happening with the determined temperature and its relationship with the application time and the response of the tissue. In this regard, I have the following comments:

  1. The document requires a complete revision of the writing, as well as the use of punctuation, for example, a punctuation symbol is missing on line 45, and line 75.

We have now corrected the many punctuation errors in the manuscript. We have also corrected the English.

  1. What software, and data acquisition hardware were used by the authors to capture and record the different temperatures at a precise time?

To clarify this issue, we have added the following text: "The data collected through the computer and Microsoft Excel software was used to calculate the mean temperature rise, the range, and the standard deviation for each tooth group".

  1. What is the meaning of the horizontal separation between the dots of the same treatment condition, i.e., for continuous irrigation, there are pink dots sparced in the axis, whit almost no temperature change.

In order to observe  all of the temperature measurements, for all the samples of the same group, these dots were separated in the horizontal direction to avoid them becoming overlapped and concealed. Each dot represents the ΔT measurement of one sample. As you noticed, there was almost no temperature change in the continuous irrigation method, which indicates that the presence of the irrigation solution in the root channel prevents the temperature rise caused by the laser.

In order to clarify this point, we have added the following text to the legend: “Each circle represents the ΔT of one sample. The pink circles represent the ΔT measurements at the coronal (cementoenamel junction) region. The green circles represent the ΔTmeasurements in the middle region, while the blue circles represent the ΔT measurements in the apical region.”

  1. Figures 3, 4, and 5, show the same information as figure 2, I suggest eliminating those figures, or they could remain if there is a real reason to be there.

We have now replaced these figures with the figures prepared according to comment 7.

  1. Tables 1, 2, and 3, are pictures with very bad quality, and their font size is different, table 3 is not easy to read. Authors must construct the tables from the latex editor, or the word editor, using a font with legible size.

We have now prepared the Tables in word formate.

  1. Results shown in table 3 are not explained in the document, a very good way to explain it is by using figures that can be described in the text. Also, table 3 doesn't show the results of the segment.

Table 3 shows the statistical data where each group was compared to the other using post-hoc Tukey’s test. It is presented to show which treatment groups are statistically significant with regard to ΔT when compared to the other group. The Table thus describes the statistics of data presented in Figures 3-6 (numbers of revised version). The text refers to Table 3 in Sections 3.1 and 3.2. In order to clarify the issue we have added to Section 2.5 the following text: “The statistical analysis was done by JUSP computer software [JASP Team (2022), Version 0.16.2].“

We have also added the following text to the Table notes to describe the different parameters.

" Notes: P-value is adjusted for comparing a family of 3. Results are averaged over the levels of Measrement_Point. Mean Difference: Average ΔT of the group of interest – Average ΔT of the compared group. Standard Error (SE) = The common error factor for all three different groups. t = The sample statistical value under the T distribution. The PTukey value stands for probability and measures the likelihood that any observed difference between groups is due to chance."

  1. A figure, or figures, showing the temperature change versus the irrigation time for the different experiments is mandatory for a better explanation of the obtained results.

We have now added these figures to the manuscript (Figures 4-6 in the revised version which replace Figures 3-5 of the original manuscript).

I think that the analysis carried out, as well as the results obtained and the explanation of these, are very poor, and the document requires a lot of work, to be considered acceptable for publication. In this case, I will allow the work to be improved and suggest major comments, hoping that the work will be substantially improved in the next version.

Thank you for your advice.

Reviewer 3 Report

The article entitled “Temperature changes during Er: YAG laser activation with 2 side-firing spiral Endo Tip in root canal treatment”, results well written. This study examines the temperature changes occurring at the outer canal surface during the activation of Er:YAG laser with the Endo tip side-firing spiral tip in the root canal. The authors recommend applying the laser treatment together with an irrigant. I suggest a minor revision concerning to implement the final part of the introduction in which the authors explain their propose.  Increased emphasis and details are recommended. The bibliography should be increased.

Author Response

We thank the reviewer for critically reading and commenting on our manuscript.

Extensive English editing is required.

The English has now been edited.

The article entitled "Temperature changes during Er: YAG laser activation with 2 side-firing spiral Endo Tip in root canal treatment", results well written. This study examines the temperature changes occurring at the outer canal surface during the activation of Er:YAG laser with the Endo tip side-firing spiral tip in the root canal. The authors recommend applying the laser treatment together with an irrigant. I suggest a minor revision concerning to implement the final part of the introduction in which the authors explain their propose.  Increased emphasis and details are recommended. The bibliography should be increased.

We have accordingly added the following text to the Introduction with the respective references:

In 1983, Eriksson and Albrektsson determined a temperature of 47 °C as the critical limit for bone survival in rabbits [‎22]. In later studies, a temperature rise of 10 °C above body temperature was found to be the critical limit [‎23]. According to a 2001 dissertation by Mazaheri in BioMed Research International 7 at RWTH Aachen College, the maximum average temperature occurring during irradiation of root canals with a 3W diode laser was below the critical limit if the optical fiber was continuously moved in a circular motion both coronally and apically [‎24]. Gutknecht et al.found that at a 3 W CW setting, the bacterial reduction was observed in bovine teeth at a depth of 500 microns [‎25]. However, the temperature limit was exceeded at 4 W with an irradiation time of 15 seconds, resulting in thermal damage [‎25].

Gutknecht observed that using lasers below 1 W is irrelevant in endodontics because they do not completely remove the smear layer or seal the dentinal tubules [‎26]. However, at settings of 1.25 W to 1.5 W, the organic material is entirely removed, and the surface of the inorganic substance is fused [‎26]. These observations are important as they help define the conditions required for preventing reinfection in the root canal.

  1. Eriksson, A.R.; Albrektsson, T. Temperature threshold levels for heat-induced hone tissue injury: a vital microscopic injury: a vital microscope study in the rabbit. Prosth. Dent. 1983; 50, 101-107. [CrossRef] [PubMed]
  2. Lan, W.H. Temperature elevation on the root surface during Nd:YAG laser irradiation in the root canal. Endod. 1999; 25(3), 155–156. [CrossRef]
  3. Mazaheri, P. Temperaturentwicklung auf der Wurzeloberfläche bei einer endodonitischen Behandlung mit einem Diodenlaser. Mainz; 2001. [Google Scholar]
  4. Gutknecht, N.; Van Gogswaardt, D.; Conrads, G.; Apel, C.; Schubert, C.; Lampert, F. Diode laser radiation and its bactericidal effect in root canal wall dentin. Clin. Laser Med. Surg. 2000; 18(2), 57–60. [CrossRef] [PubMed]
  5. Gutknecht, N. Irradiation of infected root canals with Nd:YAG lasers. A review. LaserZahnheilkunde. 2004; 4(4), 219–226.

Reviewer 4 Report

The paper demonstrates an interesting application of solid-state lasers. I believe that it is well written, explained and easy to follow. 

I think it would be convenient if the authors included a schematic of the Endo Tip when they describe it in lines 80-86. They do mention reference 20 when the authors talk about it. I looked at the reference expecting to find more about the Endo Tip, but I was not able to find more information about it. Finally, it is confusing when the authors mention the safety temperature in line 323, they mention 10C but then 46C in in brackets. 

Author Response

We thank the reviewer for critically reading and commenting on our manuscript.

The paper demonstrates an interesting application of solid-state lasers. I believe that it is well written, explained and easy to follow. 

  1. I think it would be convenient if the authors included a schematic of the Endo Tip when they describe it in lines 80-86. They do mention reference 20 when the authors talk about it. I looked at the reference expecting to find more about the Endo Tip, but I was not able to find more information about it.

We have now added the image and a schematic illustration of the Endo tip to the Introduction section (new Figure 1).

  1. Finally, it is confusing when the authors mention the safety temperature in line 323, they mention 10C but then 46C in in brackets. 

We have reformulated the sentence to make it clear.

Round 2

Reviewer 2 Report

The authors have made the suggested changes and answered the questions satisfactorily, I consider that the work can be published.